# Kernel Based Progressive Distillation for Adder Neural Networks

**Yixing Xu[1], Chang Xu[2], Xinghao Chen[1], Wei Zhang[1], Chunjing Xu[1], Yunhe Wang*[1]**
[1]Noah's Ark Lab, Huawei Technologies
[2]The University of Sydney
{yixing.xu, xinghao.chen, wz.zhang, xuchunjing, yunhe.wang}@huawei.com
c.xu@sydney.edu.au

## Abstract

Adder Neural Networks (ANNs) which only contain additions bring us a new way of developing deep neural networks with low energy consumption. Unfortunately, there is an accuracy drop when replacing all convolution filters by adder filters. The main reason here is the optimization difficulty of ANNs using $\ell_1$-norm, in which the estimation of gradient in back propagation is inaccurate. In this paper, we present a novel method for further improving the performance of ANNs without increasing the trainable parameters via a progressive kernel based knowledge distillation (PKKD) method. A convolutional neural network (CNN) with the same architecture is simultaneously initialized and trained as a teacher network, features and weights of ANN and CNN will be transformed to a new space to eliminate the accuracy drop. The similarity is conducted in a higher-dimensional space to disentangle the difference of their distributions using a kernel based method. Finally, the desired ANN is learned based on the information from both the ground-truth and teacher, progressively. The effectiveness of the proposed method for learning ANN with higher performance is then well-verified on several benchmarks. For instance, the ANN-50 trained using the proposed PKKD method obtains a 76.8% top-1 accuracy on ImageNet dataset, which is 0.6% higher than that of the ResNet-50.

## 1 Introduction

Convolutional neural networks (CNNs) with a large number of learnable parameters and massive multiplications have shown extraordinary performance on computer vision tasks, such as image classification [18, 37, 23, 17, 15], image generation [8, 7], object detection [30], semantic segmentation [26], low-level image tasks [33, 45, 35, 31, 36, 42], *etc.* However, the huge energy consumption brought by these multiplications limits the deployment of CNNs on portable devices such as cell phones and video cameras. Thus, the recent research on deep learning tends to explore efficient method for reducing the computational costs required by convolutional neural networks [47, 4, 34, 38].

There are several existing algorithms presented to derive computational-efficient deep neural networks. For example, weight pruning methods [24, 47, 20, 19, 22] removed unimportant parameters or filters from a pre-trained neural network with negligible loss of accuracy. Knowledge Distillation (KD) methods [12, 5, 3] directly learned a student model by imitating the output distribution of a teacher model. Tensor decomposition methods [28, 43, 40] decomposed the model tensor into several low-rank tensors in order to speed up the inference time.

Another direction for developing efficient neural networks is to reduce the bit of weights and activations to save the memory usage and energy consumption [9]. Hubara *et.al.* [14] proposed a

---

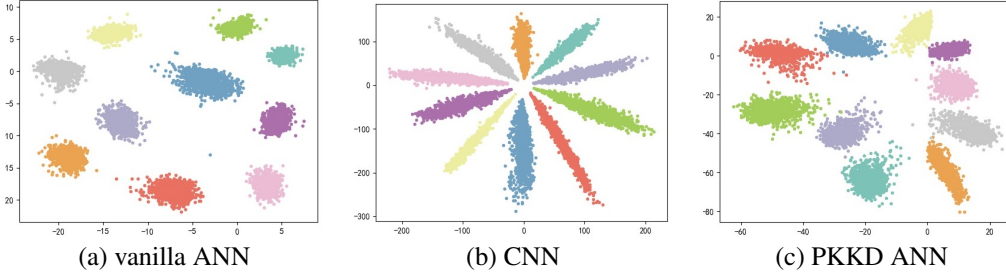

(a) vanilla ANN                    (b) CNN                    (c) PKKD ANN

Figure 1: Visualization of features in different neural networks on MNIST dataset. From left to right are vanilla ANN, CNN and PKKD ANN, respectively.

BinaryNet with both 1-bit weights and activations, which reduced multiply-accumulate operations into accumulations. Rastegari *et.al.* [29] further introduced a scale factor for each channel of weights and activations. Zhou *et.al.* [46] used a layer-wise scale factor and accelerated the training of binarized neural networks (BNNs) with low-bit gradient. Lin *et.al.* [21] used more weight and activation bases to achieve higher performance. Although these methods greatly reduce the computational complexity, the performance of resulting networks is still worse than their CNN counterparts.

Beyond the low-bit computations, Chen *et.al.* [4] proposed Adder Neural Network (ANN), which replaced the convolutional operation by using $\ell_1$-distance between activation and filter instead of correlation, thus produced a series of ANNs without massive floating number multiplications. The ANN can achieve better performance (*e.g.*, 74.9% top-1 accuracy on ImageNet with ANN-50) than low-bit especially binary neural networks, and has implications in the design of future hardware accelerators for deep learning. Thus, we are focusing on establishing ANNs with higher performance, which is comparable with that of CNNs.

To this end, we suggest to utilize a convolutional network with the same architecture and number of learnable parameters to help the training of the student adder network. Since the weights in ANN are Laplacian distribution while those in CNN are often Gaussian distribution [4], directly matching the feature information is very difficult. Thus, we develop a kernel based method for mapping features and weights of these two kinds of neural networks into a suitable space for seeking the consistency. In practice, a Gaussian alike kernel is utilized for CNN and a Laplacian kernel is used for ANN to convert their features and weights to a new space. Then, the knowledge distillation technique is applied for transferring useful information (*e.g.*, the relationship between samples and distributions) from the teacher to the student network. Moreover, instead of using a fixed teacher network, a progressive distillation method is exploited to guide the training of ANN. Experiments conducted on several benchmark models and datasets demonstrate that the proposed method can significantly improve the performance of ANNs compared to the vanilla ones, and even surpass that of CNN baselines.

## 2 Preliminaries and Motivation

In this section, we revisit ANN which uses additions to replace multiplications, and knowledge distillation (KD) method for learning student networks.

**Adder Neural Networks (ANNs).** Denote an input feature map of the intermediate layer of a deep neural network as $X \in \mathbb{R}^{H \times W \times c_{in}}$, in which $H$ and $W$ are the height and width of the feature map, and $c_{in}$ is the number of input channels. Also given a filter $F \in \mathbb{R}^{d \times d \times c_{in} \times c_{out}}$ in which $d$ is the kernel size and $c_{out}$ is the number of output channels, respectively. The traditional convolutional operation is defined as:

$$Y(u,v,c) \triangleq X * F = \sum_{i=1}^{d}\sum_{j=1}^{d}\sum_{k=1}^{c_{in}} X(u+i,v+j,k) \times F(i,j,k,c). \qquad (1)$$

To avoid massive floating number multiplications, Chen *et.al.* [4] proposed an ANN which maximizes the use of additions by replacing the convolutional operation with the $\ell_1$-distance between the input

feature map and filter:

$$\tilde{Y}(u,v,c) \triangleq X \odot F = -\sum_{i=1}^{d}\sum_{j=1}^{d}\sum_{k=1}^{c_{in}}|X(u+i,v+j,k) - F(i,j,k,c)|. \qquad (2)$$

Since Eq. 2 only contains additions, the computational and energy costs can be significantly reduced according to [6, 32, 4]. However, there is still an accuracy gap between CNN and ANN. For example, there is over 1% top-1 accuracy gap between ResNet-50 and the homogeneous ANN-50.

**Knowledge Distillation (KD).** To enhance the performance of portable student networks, KD based methods [12, 44, 11] are proposed to inherit the excellent performance from the teacher network. Given a pre-trained teacher network $\mathcal{N}_t$ and a portable student network $\mathcal{N}_s$, the student network is optimized with the following loss function:

$$\mathcal{L}_{kd} = \sum_{i=1}^{n} \mathcal{H}_{cross}(\mathbf{y}_s, \mathbf{y}_t), \qquad (3)$$

where $\mathcal{H}_{cross}$ is the cross-entropy loss, $n$ is the number of training samples, $\mathbf{y}_s = \{\mathbf{y}_s^i\}_{i=1}^n$ and $\mathbf{y}_t = \{\mathbf{y}_t^i\}_{i=1}^n$ are the outputs of student model and teacher model, respectively.

Basically, the conventional KD methods utilize a soft target to blend together the output of teacher network and ground-truth prediction:

$$\mathcal{L}_{blend} = \sum_{i=1}^{n} \big\{ \alpha \mathcal{H}_{cross}(\mathbf{y}_s, \mathbf{y}_t) + \mathcal{H}_{cross}(\mathbf{y}_s, \mathbf{y}_{gt}) \big\}, \qquad (4)$$

where $\mathbf{y}_{gt}$ is the ground-truth label, and $\alpha$ is the trade-off parameter.

Although ANN is designed with the same architecture and number of parameters with CNN, it still can be treated as a student network with lower performance. Meanwhile, this isomorphic setting is conductive to the knowledge transferred on each layer. Thus, we are motivated to explore a KD method to obtain ANNs with the same performance as (or even exceed) that of baseline CNNs.

## 3 Progressive Kernel Based Knowledge Distillation

In this section, we introduce a novel kernel based knowledge distillation method for transferring information from CNNs to ANNs. Moreover, we also investigate the progressive distillation approach for better performance.

### 3.1 Kernel Based Feature Distillation

Since ANNs are designed with the same architectures as that of CNNs, we can distill the feature maps of all intermediate layers between ANNs and CNNs. Most of the previous KD methods [41, 44] adopted a two step approach to distill the feature, *i.e.*, $1 \times 1$ convolution is firstly applied to match the feature dimension between teacher and student networks, and then the mean squared error (MSE) loss is computed based on the transformed features.

However, such two-step approach is problematic when directly used to distill the features of ANN, because the feature map distributions of ANN and CNN are different. As pointed out in [4], the weight distribution in a well-trained vanilla ANN is Laplacian distribution, while that in a well-trained vanilla CNN is Gaussian distribution. Moreover, the operation in CNN calculates the cross correlation between input and filter, and ANN calculates the $\ell_1$-distance. Denoting the input distributions of ANN and CNN as $i_a(x)$ and $i_c(x)$, and the weight distributions as $w_a(x) \sim \mathcal{L}_a(0, \lambda)$ and $w_c(x) \sim \mathcal{N}(0, \sigma^2)$, respectively. Wherein, $\lambda$ and $\sigma$ are parameters in Laplacian distribution and Gaussian distribution, the output distributions of ANN and CNN can be derived as:

$$i_a(x) \odot w_a(x) = \int_{-\infty}^{\infty} |i_a(t) - w_a(x-t)|dt = \int_{-\infty}^{\infty} |i_a(t) - \frac{1}{2\lambda}e^{-\frac{|x-t|}{\lambda}}|dt, \qquad (5)$$

and

$$i_c(x) * w_c(x) = \int_{-\infty}^{\infty} i_c(t) \times w_c(x-t)dt = \int_{-\infty}^{\infty} i_c(t) * \frac{1}{\sqrt{2\pi}\sigma}e^{-\frac{(x-t)^2}{2\sigma^2}}dt, \qquad (6)$$

respectively. By comparing Eq. 5 and Eq. 6, we can find that distributions of ANN and CNN are unlikely to be the same unless the input distributions are carefully designed. Thus, it is very hard to directly applying MSE loss to make the output feature maps similar.

Actually, the difference between the output feature maps of the two kinds of networks is caused by the distribution difference of inputs and weights. Thus, we use a kernel based method to map the inputs and weights to a higher dimensional space to alleviate this problem.

Given two different vectors $\mathbf{x}$ and $\mathbf{x}'$, a Gaussian kernel is defined as $k(\mathbf{x}, \mathbf{x}') = \exp(-\frac{||\mathbf{x}-\mathbf{x}'||^2}{2\sigma^2})$. The kernel trick non-linearly maps the input space of $\mathbf{x}$ and $\mathbf{x}'$ to a higher dimensional feature space. Motivated by this idea, given $\{\mathbf{x}_a^m, \mathbf{w}_a^m\}_{m=1}^M$ as the inputs and weights of the $m$-th layer of ANN in which $M$ is the total number of intermediate layers, and $\{\mathbf{x}_c^m, \mathbf{w}_c^m\}_{m=1}^M$ are that of CNN, respectively. We transform the output feature maps $\mathbf{x}_c^m * \mathbf{f}_c^m$ and $\mathbf{x}_a^m \odot \mathbf{f}_a^m$ using the following equation:

$$h(\mathbf{x}_c^m, \mathbf{f}_c^m) = e^{-\frac{\mathbf{x}_c^m * \mathbf{f}_c^m}{2\sigma_c^2}}, \tag{7}$$

and

$$g(\mathbf{x}_a^m, \mathbf{f}_a^m) = e^{-\frac{\mathbf{x}_a^m \odot \mathbf{f}_a^m}{\sigma_a}} \tag{8}$$

wherein, $\sigma_a$ and $\sigma_c$ are two learnable parameters. Note that for a specific point in output feature, the output of convolutional operation equals to the dot product of two input vectors, and the output of adder operation equals to the $\ell_1$-norm of two input vectors. Thus, different from the $\ell_2$-norm used in traditional Gaussian kernel, $h(\cdot)$ is a Gaussian alike kernel that computes the cross correlation, and $g(\cdot)$ is a standard Laplace kernel [27, 1] that computes the $\ell_1$-norm of two vectors. Although $h(\cdot)$ is different from the standard Gaussian kernel, we prove that it can still map the input and weight to a higher dimensional feature space (the proof is shown in the supplementary material).

**Theorem 1.** *Given input vector $\mathbf{x}$ and weight vector $\mathbf{f}$, the transformation function in Eq. 7 can be expressed as a linear combination of infinite kernel functions:*

$$e^{-\frac{\mathbf{x}*\mathbf{f}}{2\sigma^2}} = \sum_{n=0}^{\infty} < \Phi_n(\mathbf{x}), \Phi_n(\mathbf{f}) > . \tag{9}$$

*The kernel functions can be decomposed into the dot product of two mappings that maps the input space to an infinite feature space:*

$$\Phi_n(\mathbf{x}) = [\phi_n^1(\mathbf{x}), \phi_n^2(\mathbf{x}), \cdots, \phi_n^L(\mathbf{x})], \tag{10}$$

*in which $\sum_{i=1}^k n_{l_i} = n$, $n_{l_i} \in \mathbb{N}$ and $L = \frac{(n+k-1)!}{n!(k-1)!}$ goes to infinity when $n \to \infty$.*

After applying Eq. 7 and Eq. 8, the inputs and weights are mapped to a higher dimensional space, and the similarity between them is then calculated in the new space to derive the output feature maps.

Note that in the perspective of activation function, Eq. 7 and Eq. 8 can be treated as new activation functions besides ReLU. The advantage of using them instead of ReLU is that KD forces the output feature maps of teacher and student to be the same. However, when given the same inputs (which are the outputs derived from the former layer), the weight distributions and the calculation of the outputs are different in CNN and ANN, which means that the outputs should be different and is contradict with the purpose of KD. Compared to the piece-wise linear ReLU function, Eq. 7 and Eq. 8 smooth the output distribution when using a small $\sigma_c$ and $\sigma_a$ (similar to the temperature in KD), which still focus on the purpose of alleviating the difference of the distribution.

Besides using the kernel, a linear transformation $\rho(\cdot)$ is further applied to match the two distributions of the new outputs. Compare to directly using multiple layers (*e.g.*, conv-bn-relu) which is hard to train, a kernel smooths the distribution and makes the linear transformation enough to align the features. Treated ANN as an example, given $g(\mathbf{x}_a^m, \mathbf{f}_a^m)$ as the output after applying the kernel, the intermediate output used for computing KD loss is defined as:

$$\mathbf{y}_a^m = \rho(g(\mathbf{x}_a^m, \mathbf{f}_a^m); \mathbf{w}_{\rho_a}^m), \tag{11}$$

in which $\mathbf{w}_{\rho_a}$ is the parameter of the linear transformation layer. Similarly, the output of CNN is defined as:

$$\mathbf{y}_c^m = \rho(h(\mathbf{x}_c^m, \mathbf{f}_c^m); \mathbf{w}_{\rho_c}^m). \tag{12}$$

**Algorithm 1** PKKD: Progressive Kernel Based Knowledge Distillation.

---

**Input:** A CNN network $\mathcal{N}_c$, an ANN $\mathcal{N}_a$, number of intermediate layers $M$, input feature map $\mathbf{x}_a^m$, $\mathbf{x}_c^m$ and weight $\mathbf{f}_a^m$, $\mathbf{f}_c^m$ in the $m$-th layer. Training set $\{\mathcal{X}, \mathcal{Y}\}$.

1: **repeat**
2:     Randomly select a batch of data $\{x^i, y^i\}_{i=1}^n$ from $\{\mathcal{X}, \mathcal{Y}\}$, where $n$ is the batchsize;
3:     **for** $m = 1, \cdots, M$ **do**:
4:         Calculate the ANN output in the $m$-th layer $\mathbf{x}_a^m \odot \mathbf{f}_a^m$;
5:         Transform the output feature of ANN using Eq. 11 to obtain $\mathbf{y}_a^m$;
6:         Calculate the CNN output in the $l$-th layer $\mathbf{x}_c^m * \mathbf{f}_c^m$;
7:         Transform the output feature of CNN using Eq. 12 to obtain $\mathbf{y}_c^m$;
8:     **end for**
9:     Calculate the loss function $\mathcal{L}_{mid}$ in Eq. 13;
10:    Obtain the softmax outputs of CNN and ANN and denote as $\mathbf{y}_c$ and $\mathbf{y}_a$, respectively.
11:    Compute the loss function $\mathcal{L}_{blend}$ in Eq. 4;
12:    Apply the KD loss $\mathcal{L} = \beta\mathcal{L}_{mid} + \mathcal{L}_{blend}$ for $\mathcal{N}_a$;
13:    Calculate the normal cross entropy loss $\mathcal{L}_{ce} = \sum_{i=1}^n \mathcal{H}_{cross}(\mathbf{y}_c^i, \mathbf{y}^i)$ for $\mathcal{N}_c$;
14:    Update parameters in $N_c$ and $N_a$ using $\mathcal{L}_{ce}$ and $\mathcal{L}$, respectively;
15: **until** converge
**Output:** The resulting ANN $\mathcal{N}_a$ with excellent performance.

---

Note that the goal of ANN is to solve the classification problem rather than imitate the output of CNN. Thus, the feature alignment problem is solved by applying the linear transformation after using the kernel. The main stream of ANN is unchanged which means the inference is exactly the same as the vanilla ANN. Finally, the intermediate outputs of ANN is distilled based on the intermediate outputs of CNN. Specifically, a $1 \times 1$ convolutional operation is used as the linear transformation $\rho(\cdot)$ during the experiment.

The knowledge distillation is then applied on each intermediate layer except for the first and last layer. Given the intermediate outputs of ANN and CNN, the KD loss is shown below:

$$\mathcal{L}_{mid} = \sum_{i=1}^n \sum_{m=1}^M \mathcal{H}_{mse}(\mathbf{y}_a^m, \mathbf{y}_c^m). \tag{13}$$

Combining Eq. 13 and Eq. 4, the final loss function is defined as:

$$\begin{aligned}
\mathcal{L} &= \beta\mathcal{L}_{mid} + \mathcal{L}_{blend} \\
&= \sum_{i=1}^n \sum_{m=1}^M \beta\mathcal{H}_{mse}(\mathbf{y}_a^m, \mathbf{y}_c^m) + \sum_{i=1}^n \left\{ \alpha H_{cross}(\mathbf{y}_s, \mathbf{y}_t) + \mathcal{H}_{cross}(\mathbf{y}_s, \mathbf{y}_{gt}) \right\},
\end{aligned} \tag{14}$$

where $\beta$ is the hyper-parameter for seeking the balance between two loss functions.

### 3.2 Learning from a Progressive CNN

Several papers had pointed out that in some circumstances the knowledge in teacher model may not be transferred to student model well. The reasons behind are two folds.

The first is that the difference of the structure between teacher and student model is large. Prior works try to use assistant model to bridge the gap between teacher and student [25]. However, the reason why a large gap degrades the performance of student is not clear. We believe that the divergence of output distributions is the reason that cause the problem, and we map the output to a higher dimensional space to match the distributions between teacher and student, as shown in previous.

The second is that the divergence of training stage between teacher model and student model is large. Thus, researches have focused on learning from a progressive teacher. Jin *et.al.* [16] required an anchor points set from a series of pre-trained teacher networks, and the student is learned progressively from the anchor points set. Yang *et.al.* [39] assumed that teacher and student have the same architecture, and the student is learned from the teacher whose signal is derived from an earlier iteration. Methods mentioned above require storing several pre-trained teacher models, and are memory-consuming.

Table 1: Impact of different components of the proposed distillation method.

| | 1 | 2 | 3 | 4 | 5 | 6 | 7 | 8 |
|---|---|---|---|---|---|---|---|---|
| KD Loss | ✓ | | | | ✓ | | | |
| $1 \times 1$ conv + KD Loss | | ✓ | | | | ✓ | | |
| Kernel + KD Loss | | | ✓ | | | | ✓ | |
| Kernel + $1 \times 1$ conv + KD Loss | | | | ✓ | | | | ✓ |
| Fixed teacher | ✓ | ✓ | ✓ | ✓ | | | | |
| Progressively learned teacher | | | | | ✓ | ✓ | ✓ | ✓ |
| Accuracy(%) | 92.21 | 92.27 | 92.29 | 92.39 | 92.58 | 92.67 | 92.75 | **92.96** |

We believe that the second circumstance is also related to the distribution discrepancy between the teacher and student model. In this time, it is the difference of training stage rather than the difference of architecture that makes the distribution different. Thus, in this section we move a step further and use a more greedy manner by simultaneously learning the CNN and ANN. Given a batch of input data, CNN is learned normally using the cross-entropy loss, and ANN is learned with the KD loss by using the current weight of CNN, *i.e.*

$$\mathcal{L}^b = \beta \mathcal{L}^b_{mid} + \mathcal{L}^b_{blend}, \tag{15}$$

in which $b$ denotes the current number of step. When doing back-propagation, the KD loss is only backprop through ANN, and CNN is learned without interference.

There are two benefits by using this way. Firstly, only one copy of teacher model is required during the whole training process, which drastically reduces the memory cost compared to the previous methods, and is equal to the conventional KD method. Secondly, to the maximum extent it alleviates the effect of different training stage of CNN and ANN that brings the discrepancy of output distributions. The proposed progressive kernel based KD method (PKKD) is summarized in Algorithm 1.

The benefit of the proposed PKKD algorithm is explicitly shown in Figure 1. In CNN, the features can be classified based on their angles, since the convolution operation can be seen as the cosine distance between inputs and filters when they are both normalized. The features of vanilla ANN are gathered together into clusters since $\ell_1$-norm is used as the similarity measurement. ANN trained with the proposed PKKD algorithm combines the advantages of both CNN and ANN, and the features are gathered together while at the same time can be distinguished based on their angles.

## 4 Experiments

In this section, we conduct experiments on several computer vision benchmark datasets, including CIFAR-10, CIFAR-100 and ImageNet.

### 4.1 Experiments on CIFAR

We first validate our method on CIFAR-10 and CIFAR-100 dataset. CIFAR-10 (CIFAR-100) dataset is composed of $50k$ different $32 \times 32$ training images and $10k$ test images from 10 (100) categories. A commonly used data augmentation and data pre-processing method is applied to the training images and test images. An initial learning rate of $0.1$ is set to both CNN and ANN, and a cosine learning rate scheduler is used in training. Both models are trained for 400 epochs with a batchsize of 256. During the experiment we set hyper-parameters $\alpha = \beta \in \{0.1, 0.5, 1, 5, 10\}$, and the best result among them is picked.

In the following we do an ablation study to test the effectiveness of using kernel based feature distillation and a progressive CNN during the learning process. Specifically, the ablation study is conducted on CIFAR-10 dataset. The teacher network is ResNet-20, and student network uses the same structure except that the convolutional operations (Eq. 1) are replaced as adder operations (Eq. 2) to form an ANN-20. The first and last layers are remain unchanged as in [4].

Four different settings are conducted to prove the usefulness of kernel based feature distillation. The first setting is directly computing the KD loss on the output of intermediate layers of ResNet-20 and ANN-20, the second is computing the KD loss after applying a $1 \times 1$ convolution to the output of the features, which is commonly used in many previous research [44, 11]. The third is using the kernel

Table 2: Classification results on CIFAR-10 and CIFAR-100 datasets.

| Model | Method | #Mul. | #Add. | XNOR | CIFAR-10 | CIFAR-100 |
|---|---|---|---|---|---|---|
| VGG-small | CNN | 0.65G | 0.65G | 0 | 94.25% | 75.96% |
| | BNN | 0.05G | 0.65G | 0.60G | 89.80% | 67.24% |
| | ANN | 0.05G | 1.25G | 0 | 93.72% | 74.58% |
| | MMD ANN [13] | 0.05G | 1.25G | 0 | 93.97% | 75.14% |
| | PKKD ANN | 0.05G | 1.25G | 0 | **95.03%** | **76.94%** |
| ResNet-20 | CNN | 41.17M | 41.17M | 0 | 92.93% | 68.75% |
| | BNN | 0.45M | 41.17M | 40.72M | 84.87% | 54.14% |
| | ANN | 0.45M | 81.89M | 0 | 92.02% | 67.60% |
| | MMD ANN [13] | 0.45M | 81.89M | 0 | 92.30% | 68.07% |
| | PKKD ANN | 0.45M | 81.89M | 0 | **92.96%** | **69.93%** |
| ResNet-32 | CNN | 69.12M | 69.12M | 0 | 93.59% | 70.46% |
| | BNN | 0.45M | 69.12M | 68.67M | 86.74% | 56.21% |
| | ANN | 0.45M | 137.79M | 0 | 93.01% | 69.17% |
| | MMD ANN [13] | 0.45M | 137.79M | 0 | 93.16% | 69.89% |
| | PKKD ANN | 0.45M | 137.79M | 0 | **93.62%** | **72.41%** |

without linear transformation, and the fourth is the proposed method mentioned above. The other parts are the same for the above four settings.

In order to verify the effectiveness of using a progressive teacher during training, a pre-trained fixed ResNet-20 and a progressively learned ResNet-20 model are used separately. Thus, there are a total of 8 different settings in the ablation study. The experimental results are shown in Table 1.

The results show that using a progressively learned teacher has a positive effect on the knowledge distillation in all the circumstances. Directly applying $1 \times 1$ convolutional operation benefits to the knowledge transfer, but the divergence of the output distribution cannot be eliminated with such linear transformation. In fact, the combination of using a kernel based transformation and a $1 \times 1$ convolution performs best among the four different settings. After all, combining the kernel based feature distillation method and the usage of progressive CNN, we get the best result of $92.96\%$ on CIFAR-10 with ANN-20.

In the following, we use the best setting mentioned above for other experiments. The VGG-small model [2], ResNet-20 model and ResNet-32 model are used as teacher models, and the homogeneous ANNs are used as student models. The vanilla ANN [4], the binary neural network (BNN) using XNOR operations instead of multiplications [46] and MMD [13] method are used as the competitive methods. MMD mapped the outputs of teacher and student to a new space using the same kernel function, which is different from ours that maps the inputs and weights to a higher dimensional space and using different kernel functions for ANN and CNN. Note that by applying Gaussian alike kernel for CNN and Laplace kernel for ANN, our method is able to directly alleviate the problem that the weight distributions are different in ANN and CNN by separately mapping them to new space using the corresponding kernels. As the results shown in Table 2, the proposed method achieves much better results than vanilla ANN, and even outperforms the homogeneous CNN model. On VGG-small model, PKKD ANN achieves 95.03% accuracy on CIFAR-10 and 76.94% accuracy on CIFAR-100, which is 0.78% and 0.98% better than CNN. On the widely used ResNet model, the conclusion remains the same. For ResNet-20, PKKD ANNs achieves the highest accuracy with 92.96% on CIFAR-10 and 69.93% on CIFAR-100, which is 0.03% and 1.18% higher than the homogeneous CNN. The results on ResNet-32 also support the conclusion.

We further report the training and testing accuracy of ResNet-20, vanilla ANN-20 and the PKKD ANN-20 models on CIFAR-10 and CIFAR-100 datasets to explicitly get an insight of the reason why the PKKD ANN derived from the proposed method performs even better than the homogeneous CNN model. In Figure 2, the solid lines represent the training accuracy and the dash lines represent the testing accuracy. On both CIFAR-10 and CIFAR-100 datasets, the CNN model achieves a higher training and testing accuracy than the vanilla ANN model. This is because when computing the gradient in vanilla ANN, the derivative of $\ell_2$-norm is used instead of the original derivative [4], and the loss may not go through the correct direction. When using the proposed PKKD method, the output of CNN is used to guide the training of ANN, thus lead to a better classification result. Moreover,

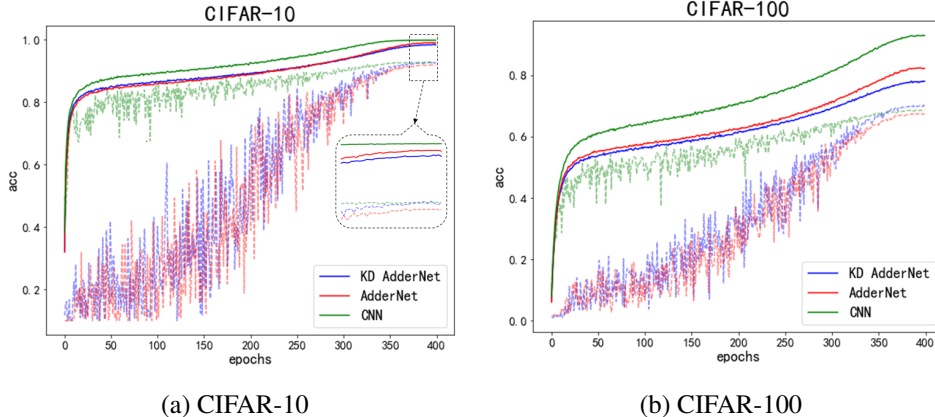

| | (a) CIFAR-10 | | | | (b) CIFAR-100 |

Figure 2: Training and testing accuracy of ResNet-20, vanilla ANN-20 and the proposed PKKD ANN-20 models on CIFAR-10 and CIFAR-100 datasets.

Table 3: Experimental results of using different hyper-parameters.

| $\alpha/\beta$ | 0.1 | 0.5 | 1 | 5 | 10 |
|---|---|---|---|---|---|
| CIFAR-10 | 92.49% | 92.58% | 92.61% | 92.78% | **92.96%** |
| CIFAR-100 | 68.51% | 68.73% | 69.15% | **69.93%** | 69.62% |

applying KD method helps preventing the student network from over-fitting [12], which is the reason why PKKD ANN has the lowest training accuracy and highest testing accuracy.

Finally, we show the classification results of using different hyper-parameters $\alpha$ and $\beta$. We set hyper-parameters $\alpha = \beta \in \{0.1, 0.5, 1, 5, 10\}$, the teacher network is ResNet-20, and the student network is the homogeneous ANN-20. Experiments are conducted on CIFAR-10 and CIFAR-100 datasets and the results are shown in Tab. 3. More experiments compared with other methods are shown in the supplementary material.

## 4.2 Experiments on ImageNet

We also conduct experiments on ImageNet dataset. The dataset is composed of over $1.2M$ different $224 \times 224$ training images and $50k$ test images from 1000 different categories. The same data augmentation and data pre-processing method is used as in He *et.al.* [10]. ResNet-18 and ResNet-50 are used as teacher models, and the homogeneous ANNs are used as student models. The teacher and student models are trained for 150 epochs with an initial learning rate of 0.1 and a cosine learning rate decay scheduler. The weight decay and momentum are set to $0.0001$ and $0.9$, respectively. The batchsize is set to 256, and the experiments are conducted on 8 NVIDIA Tesla V100 GPUs.

As the results shown in Table 4, ANN trained with the proposed PKKD method on ResNet-18 achieves a 68.8% top-1 accuracy and 88.6% top-5 accuracy, which is 1.8% and 1.0% higher than the vanilla ANN, and reduce the gap from the original CNN model. The results show again that the proposed PKKD method can extract useful knowledge from the original CNN model, and produce a comparable results with a much smaller computational cost by replacing multiplication operations with addition operations. XNOR-Net [29] tries to replace multiplication operations with xnor operations by quantifying the output, but it achieves a much lower performance with only 51.2% top-1 accuracy and 73.2% top-5 accuracy. We also report the accuracies on ResNet-50, and the conclusion remains the same. The proposed PKKD ANN achieves a 76.8% top-1 accuracy and 93.3% top-5 accuracy, which is 1.9% and 1.6% higher than the vanilla ANN, and is 0.6% and 0.4% higher than the original CNN model. Thus, we successfully bridge the gap between ANN and CNN by using the kernel based feature distillation and a progressive learned CNN to transfer the knowledge from CNN to a homogeneous ANN.

Table 4: Classification results on ImageNet.

| Model | Method | #Mul. | #Add. | XNOR | Top-1 Acc | Top-5 Acc |
|---|---|---|---|---|---|---|
| | CNN | 1.8G | 1.8G | 0 | 69.8% | 89.1% |
| | BNN | 0.1G | 1.8G | 1.7G | 51.2% | 73.2% |
| ResNet-18 | ANN | 0.1G | 3.5G | 0 | 67.0% | 87.6% |
| | MMD ANN [13] | 0.1G | 3.5G | 0 | 67.9% | 88.0% |
| | PKKD ANN | 0.1G | 3.5G | 0 | **68.8%** | **88.6%** |
| | CNN | 3.9G | 3.9G | 0 | 76.2% | 92.9% |
| | BNN | 0.1G | 3.9G | 3.8G | 55.8% | 78.4% |
| ResNet-50 | ANN | 0.1G | 7.6G | 0 | 74.9% | 91.7% |
| | MMD ANN [13] | 0.1G | 7.6G | 0 | 75.5% | 92.2% |
| | PKKD ANN | 0.1G | 7.6G | 0 | **76.8%** | **93.3%** |

## 5    Conclusion

Adder neural networks are designed for replacing massive multiplications in deep learning by additions with a performance drop. We believe that such kind of deep neural networks can significantly reduce the computational complexity, especially the energy consumption of computer vision applications. In this paper, we show that the sacrifice of performance can be compensated using a progressive kernel based knowledge distillation (PKKD) method. Specifically, we use the kernel based feature distillation to reduce the divergence of distributions caused by the difference of operations and weight distributions in CNN and ANN. We further use a progressive CNN to guide the learning of ANN to reduce the divergence of distributions caused by the difference of training stage. The experimental results on several benchmark datasets show that the proposed PKKD ANNs produce much better classification results than vanilla ANNs and even outperform the homogeneous CNNs, which make ANNs both efficient and effective.

## Broader Impact

Adder Neural Network (ANN) is a new way of generating neural networks without using multiplication operations. It will largely reduce the energy cost and the area usage of the chips. This paper makes the performance of ANN exceeded that of homogeneous CNN, which means that we can use less energy to achieve a better performance. This is beneficial to the application of smart phones, the Internet of things, *etc.*

### Funding Disclosure

We thank anonymous area chair and reviewers for their helpful comments. Chang Xu was supported by the Australian Research Council under Project DE180101438.

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
