[Supplementary Material]

# Supplementary Material:
# Progressive Kernel Based Knowledge Distillation for Adder Neural Networks

**Yixing Xu[†], Chang Xu[‡], Xinghao Chen[†], Wei Zhang[†], Chunjing Xu[†], Yunhe Wang[†]**
[†]Noah's Ark Lab, Huawei Technologies
[‡]The University of Sydney
{yixing.xu, xinghao.chen, wz.zhang, xuchunjing, yunhe.wang}@huawei.com
c.xu@sydney.edu.au

## 1   Proof of Theorem 1

Given two input vector $\mathbf{x}$ and $\mathbf{f}$, the result of convolutional operation on a specific point of the image is the dot product of two vectors. Thus, Eq.(7) in the main paper can be written as:

$$e^{-\frac{\mathbf{x}*\mathbf{f}}{2\sigma^2}} = e^{-\frac{\mathbf{x}^\top \mathbf{f}}{2\sigma^2}} = \sum_{n=0}^{\infty} \frac{(-\frac{\mathbf{x}^\top \mathbf{f}}{2\sigma^2})^n}{n!} \text{ (Taylor series expansion)}$$

$$= \sum_{n=0}^{\infty} P_n \frac{(\sum_{i=1}^{k} x_i f_i)^n}{n!} \text{ (denote } P_n = (-\frac{1}{2\sigma^2})^n ). \tag{1}$$

Given the polynomial expansion theorem $(\sum_{i=1}^{k} x_i)^n = \sum_{l=1}^{L} \frac{n!}{n_{l_1}! n_{l_2}! \cdots n_{l_k}!} x_1^{n_{l_1}} x_2^{n_{l_2}} \cdots x_k^{n_{l_k}}$, in which $\sum_{i=1}^{k} n_{l_i} = n$, $n_{l_i} \in \mathbb{N}$ and $L = \frac{(n+k-1)!}{n!(k-1)!}$, Eq.(1) can be expressed as:

$$e^{-\frac{\mathbf{x}^\top \mathbf{f}}{2\sigma^2}} = \sum_{n=0}^{\infty} P_n \frac{(\sum_{i=1}^{k} x_i f_i)^n}{n!}$$

$$= \sum_{n=0}^{\infty} P_n \frac{1}{n!} \sum_{l=1}^{L} \frac{n!}{n_{l_1}! n_{l_2}! \cdots n_{l_k}!} (x_1 f_1)^{n_{l_1}} (x_2 f_2)^{n_{l_2}} \cdots (x_k f_k)^{n_{l_k}}$$

$$= \sum_{n=0}^{\infty} \sum_{l=1}^{L} \sqrt{\frac{P_n}{n_{l_1}! n_{l_2}! \cdots n_{l_k}!}} (x_1^{n_{l_1}} x_2^{n_{l_2}} \cdots x_k^{n_{l_k}}) \sqrt{\frac{P_n}{n_{l_1}! n_{l_2}! \cdots n_{l_k}!}} (f_1^{n_{l_1}} f_2^{n_{l_2}} \cdots f_k^{n_{l_k}})$$

$$= \sum_{n=0}^{\infty} \sum_{l=1}^{L} \phi_{n_l}(\mathbf{x}) \phi_{n_l}(\mathbf{f}) \quad \text{(denote } \phi_{n_l}(\mathbf{x}) = \sqrt{\frac{P_n}{n_{l_1}! n_{l_2}! \cdots n_{l_k}!}} (x_1^{n_{l_1}} x_2^{n_{l_2}} \cdots x_k^{n_{l_k}}))$$

$$= \sum_{n=0}^{\infty} <\Phi_n(\mathbf{x}), \Phi_n(\mathbf{f})> \quad \text{(denote } \Phi_n(\mathbf{x}) = [\phi_n^1(\mathbf{x}), \phi_n^2(\mathbf{x}), \cdots, \phi_n^L(\mathbf{x})])$$

$$= \sum_{n=0}^{\infty} \mathcal{K}_n(\mathbf{x}, \mathbf{f}). \tag{2}$$

Thus, the transformation in Eq.(7) in the main paper can be expressed as a linear combination of infinite kernel functions, which means the output space is mapped to an infinite dimensional space. Also note that when $n \to \infty$, $L$ also goes to infinity, which means that the input space is mapped to an infinite dimensional space.

## 2 More Experimental Results of PKKD

In this section, more experimental results of PKKD are conducted. We compared the proposed method with other methods, such as ANN+dropout, Snapshot-KD [3], SP-KD [2], Gift-KD [4] and AT [5] on ResNet-20 using CIFAR-10 dataset as shown in Tab. 1.

Table 1: Compared with other methods on ResNet-20 using CIFAR-10 dataset.

| PKKD | ANN + dropout | Snapshot-KD [3] | SP-KD [2] | Gift-KD [4] | AT [5] |
|---|---|---|---|---|---|
| **92.96%** | 92.20% | 92.33% | 92.38% | 92.22% | 92.27% |

Then, we show the superiority of the proposed methods on the traditional CNN distillation. We compared the proposed method with vanilla KD [1] on ImageNet dataset using ResNet-152 as teacher model and ResNet-18 as student model. The results are shown in Tab. 2.

Table 2: PKKD and KD in CNN distillation.

| Model | Top-1 acc | Top-5 acc |
|---|---|---|
| ResNet-18 | 69.8% | 89.1% |
| PKKD | **73.1%** | **91.3%** |
| Vanilla KD [1] | 72.5% | 90.9% |

Finally, we show the experimental results of using different settings of PKKD on ImageNet with ResNet-50 in Tab. 3.

Table 3: Ablation study on ImageNet with ResNet-50. 'K / NK' stands for using kernel or not. 'P / NP' stands for using progressive or fixed teacher.

| CNN | ANN | K + P (PKKD) | NK + P | K + NP | NK + NP |
|---|---|---|---|---|---|
| 76.2% / 92.9% | 74.9% / 91.7% | **76.8% / 93.3%** | 75.9% / 92.6% | 75.6% / 92.2% | 75.2% / 92.0% |