[Reviews · NeurIPS 2020]

Review 1

Summary and Contributions: This paper proposes a novel way to increase the accuracy of ANN, which is a new deep learning architecture without using multiplications. It solves the problem by integrating kernel method into knowledge distillation. The teacher model is randomly initialized and simultaneously learned with the student model. The experimental results demonstrate the usefulness of the paper and ANN achieves better accuracy than CNN, which is extremely important when applying ANN to the real-world applications.

Strengths: The motivation is clear. The reason of using kernel based knowledge distillation method and using unfixed teacher with random initialization is clearly stated. Both ways are used to reduce the diversity of distributions between CNN and ANN. The usefulness of the idea is verified by the ablation study in the experiment. A pivotal theoretical proof. The paper first show the output distribution of ANN and CNN in Eq.5 and 6 to demonstrate the diversity of the two distributions. Then, the kernel method is proved to map the original input space to an infinite dimensional feature space, which convince people that the kernel method can reduce the diversity of output distributions. Novelty of the proposed method. The paper applies two different kernels to the inputs and weights of ANN and CNN, rather than a single kernel to the outputs of the two networks. As the paper points out, the divergence of the distributions is caused by the different ‘ground-truth’ weight distribution and different operations. Thus, this novel method essentially solves the problem. Extensive experiments. The ablation study well demonstrate the usefulness of each part of the proposed method. Experimental results on small dataset CIFAR and large scale dataset ImageNet both show the priority of the proposed method.

Weaknesses: The paper claims that the proposed method has a better generalization probability in Figure 2. I wonder whether the proposed method can be replaced by other generalization methods, such as dropout. The authors may compare their method with the vanilla ANN with dropout. It is well-known that the traditional KD method does not change the inference process. Does the kernel based method influence the inference process? If so, there is still a lot of multiplications during inference. The proposed kernel based method gives impressive experimental results on several benchmark datasets. Is it possible to apply this method to the traditional CNN distillation methods? When the two networks have a huge divergence of capacity (e.g. ResNet-152 and ResNet-18), the standard KD method does not work well. If this phenomenon is also caused by the ‘divergence of output distributions’, it may also be solved with the proposed method. The author may conduct experiments on this setting.

Correctness: Yes.

Clarity: The paper is well written and easy to follow.

Relation to Prior Work: Yes.

Reproducibility: Yes

Additional Feedback: see weakness. ------ Post Rebuttal ------ I have read the rebuttal and keep my prior rating. The authors addressed my concern and I recommend acceptance.


Review 2

Summary and Contributions: This paper produces a novel kernel-based progressive distillation method to improve the accuracy of the Adder Neural Network with the help of CNN. They applies kernel functions to the output feature maps of middle layers to mitigate the diversity of the weight distribution in ANN and CNN. They use a progressively learned CNN rather than a fixed one to help the learning of ANN to improve the accuracy. Compared to the previous methods, the proposed method only needs to store a single copy of the teacher model, which saves the memory. The accuracy of ANN learned with the proposed method surpass the original ANN by a margin, and also surpass the CNN network. The results may contribute to the deep learning society, since it shows that deep neural networks can achieve better results with much less multiplication compared to CNN. This paper encourages us that CNN can be replaced by ANN without loss of performance, which means that less computational resource is required to achieve the same classification accuracy. I believe that this paper contributes a lot to the application of ANN.

Strengths: Overall, this paper is clearly written and easy to understand. The paper proposes a novel way of embedding the kernel method to the knowledge distillation method, and improve the progressive KD method by storing only one copy of the teacher model. The technical part is quite solid. The paper points out that the ineffectiveness of the traditional KD method is caused by the difference of the output distribution, and further conclude two reasons that cause the problem. The first is that the weight distributions and the operations are different in CNN and ANN, and the second is that the training stages are different. The solution of using kernel and progressive teacher is reasonable. They also prove the proposed kernel can map the input space into higher dimensional space that support the results. The ablation study justifies the previous analysis and shows the superiority of using kernel and progressive CNN. The experimental results are quite encouraging. The proposed PKKD ANN can surpass the results of CNN, and the authors explain the reason in the experimental analysis. The contribution of the paper is significant and novel. Although the first ANN paper [1] proposed a new way of building deep neural networks without using multiplication, there is still a performance gap between it and the traditional ANN. This paper successfully bridges the gap between ANN and CNN, which may encourage more people to use ANN in real-world applications and also encourage the design of new chips for ANN. [1] AdderNet: Do we really need multiplications in deep learning?

Weaknesses: Why do the authors use a homogeneous CNN to distill ANN, rather than using a larger ANN as the traditional knowledge distillation for CNN do? The authors are encouraged to compare their method with some self-distillation methods in the experiment, for example [2]. ------------------------------------------- I appreciate the authors' feedback. The authors' feedback well answer my concerns. I stand by my recommendation to accept the paper. The experimental results of using different alpha/beta should be placed in the main paper for integrity. [2] Snapshot Distillation: Teacher-Student Optimization in One Generation.

Correctness: Yes. The method is reasonable, and the results of ablation study justify the analysis of the method.

Clarity: Yes. This paper is clearly written and easy to understand.

Relation to Prior Work: Yes. This paper compares the proposed method with state-of-the-art previous method, and discuss the difference in the experimental analysis.

Reproducibility: Yes

Additional Feedback: Please refer to the above weaknesses.


Review 3

Summary and Contributions: The authors of this submission targets at improving the performance of a recently proposed architecture Adder Neural Networks (ANNs), using the popular knowledge distillation method. At the heart of this approach is a a kernel based knowledge distillation method to address the different feature distributions of CNNs and ANNs. Specifically, the authors adopted a progressive distillation method to learn the ANN rather than a fixed teacher network. Experiments on CIFAR-10/100 and ImageNet demonstrate that the proposed method improves the accuracy of ANN, even better performance than the CNN counterparts.

Strengths: Overall, the proposed method is well grounded and justified. Here are the strengths I read. 1. I find the method proposed here to be novel and interesting. ANNs have been proven efficient yet still suffers from performance drops. The authors propose here an interesting approach to minimizing the performance gaps between ANNs and CNNs, which will potentially be of interest to a large audience. 2. The motivation behind is clear and is justified with the experiments. The theoretical analysis seems good. 3. Learning from a progressive teacher network is innovative . Despite the fact that it introduces more training burden, the inference complexity remains the same, making it more practical. 4. The performance on CIFAR and ImageNet are quite impressive. 5. The visualization in Figure 1 give a good illustration of the proposed method.

Weaknesses: 1. As as stated in P6 line 206, the first and last layers of remain to be convolutional, In Table 2 and Table 3, however, the #Mul for ANNs are all zero. Please clarify this. 2.The method is compared with XNOR-Net [27], a 2016 approach, after which a chain of BNN models have been proposed. It would be optimal to compare to these models as well, e.g.,PCNN (Gu et. al, AAAI 2019), and PCNN (Gu et. al, AAAI 2019) 3. Please show the tradeoff, in terms of the extra training time required.

Correctness: Yes, they are correct to my best knowledge.

Clarity: Yes, well written.

Relation to Prior Work: Yes. Some prior works, mentioned in the Weakness, should also be discussed.

Reproducibility: Yes

Additional Feedback: I spotted some typos, tiny ones: -P3, line 81: kD methods utilizes -> KD methods utilize -P3, line 76: KD based method are -> KD based methods are ------ Post Rebuttal ------ I have read the rebuttal and keep my prior rating. The authors addressed my concern and I recommend acceptance.


Review 4

Summary and Contributions: This paper proposes a progressive kernel based knowledge distillation (PKKD) method for improving the performance of adder neural networks (ANN). The features of teacher convolutional neural network (CNN) and student ANN are mapped to higher dimensional reproducing kernel Hilbert Spaces for feature matching. During training, the teacher CNN is also updated using cross-entropy loss together with the student ANN for progressive learning. Experiments are conducted on CIFAR and ImageNet datasets and promising results are reported.

Strengths: The paper is clearly written. The kernel based feature distillation to reduce the distribution divergence of teacher CNN and student ANN seems reasonable, and progressive training of the teacher CNN seems to bring moderate benefit compared to commonly used fixed teacher networks. The experimental results show that the proposed knowledge distillation method improves the student ANN.

Weaknesses: The effectiveness of the kernel method, one of the claimed contributions, is not fully justified. As shown in Table 1, the kernel operation brings insignificant gain on CIFAR 10 with a shallower network of ResNet-20. Specifically, for fixed teacher, the recognition accuracies of 1x1 conv+KD Loss [R1] vs. kernel+1x1 conv+KD losss vs are 92.29% vs. 92.39%, and for progressively teacher, their respective accuracies are 92.75% vs 92.96%. The gains (below 0.21%) seems insignificant, which may be due to stochastic initialization of networks, suggesting that the proposed kernel scheme may not be so effective as advocated. I advised that comparison on ImageNet with a deeper network (e.g., ResNet-50) is performed. [R1] A. Romero, N. Ballas, S. E. Kahou, A. Chassang, C. Gatta, and Y. Bengio. FitNets: hints for thin deep nets. ICLR, 2015. The current experiments are not strong to support that the proposed method is a competitive knowledge distillation method. Comparison with the competing methods are missing, including but not limited to Tung et al. [R2], Yim et al. [25] and Zagoruyko et al. [26]. Without these comparisons, it will not be convincing to validate that proposed method is a competitive knowledge distillation method, superior to state-of-the-arts for the adder neural networks. [R2] F. Tung G. Mori. Similarity-Preserving Knowledge Distillation. ICCV, 2019. ------------------------------------------------------------------------------------------------------ I appreciate the authors' response to my comments. One of my primary concerns is that the effectiveness of proposed kernel strategy is not justified, and the corresponding performance gains is insignificant on CIFAR (less than 0.21%), compared to the method of 1x1 convolution without kernel [R1]. In the rebuttal, the authors state that their method is obtained by averaging of 5 different runs. Unfortunately, this cannot account for the small improvement I pointed out; furthermore, they fail to provide standard deviations, and so it is hard to say the small improvement has statistical significance. The additional experiments on ImageNet show that the gain of kernel strategy is not significant (0.9%) as well, i.e., K+P vs NK+P 73.4 vs 72.5. In addition, these results are obtained with partially converged networks with only 40 epochs and it is not clear whether the gains would persist if the networks converged after sufficient epochs. My second concern is comparison with state-of-the-arts. Table 3 in the rebuttal has given comparison results on CIFAR, showing that the improvement over [R1] is small (i.e., 0.58%). However, comparison results on ImageNet are still missing. In short, I am afraid the authors fail to address my concerns. Besides, the progressive scheme is of very limited novelty, as it consists in joint updating of the teacher and student that is widely used in knowledge distillation works (e.g. [24]). As such, I keep my preliminary recommendation unchanged. [R1] A. Romero, N. Ballas, S. E. Kahou, A. Chassang, C. Gatta, and Y. Bengio. FitNets: hints for thin deep nets. ICLR, 2015.

Correctness: The proposed method seems to be technically correct.

Clarity: The paper is clearly written and easy to follow.

Relation to Prior Work: The following work should be referenced, in which convolutional layers are first applied before features matching by mean square error (MSE) between teacher and student networks. Differences from this work should be explicitly clarified. [R1] A. Romero, N. Ballas, S. E. Kahou, A. Chassang, C. Gatta, and Y. Bengio. FitNets: hints for thin deep nets. ICLR, 2015.

Reproducibility: Yes

Additional Feedback:

[Author Response · NeurIPS 2020]

We sincerely thank the anonymous reviewers for their supports and constructive comments.

**Response to R1:**

*Q1: Compare with vanilla ANN with dropout.* A1: The classification accuracy is improved not only because of the better generalization ability. The gradient used in vanilla ANN is L2-gradient rather than L1-gradient which is not accurate. The knowledge from CNN will help ANN to find the correct optimization direction. Only adding dropout to vanilla ANN will not solve this problem. The experimental results on CIFAR-10 are shown in Table 1 column 2 and 4.

*Q2: Does PKKD change the inference process?* A2: As shown in Line-150 in the main paper, the inference process is exactly the same as the vanilla ANN, which means that the proposed method will not introduce extra multiplication operations during inference. We will further emphasize this in the final version.

*Q3: Apply the method to the traditional CNN distillation methods.* A3: It is really a good question. We believe that the proposed PKKD method is also useful for the distillation between traditional CNNs. When the architectures of the two networks are different, their 'ground-truth' weight distributions and feature maps of intermediate layers are also different. The experimental results of distilling ResNet-18 with ResNet-152 are shown in Table 4.

**Response to R2:**

*Q1: Using CNN rather than bigger ANN to distill.* A1: Theoretically we can use bigger ANN to distill smaller ANN. However, it is not always possible to find a bigger teacher. Instead, we can always find a homogeneous CNN as teacher model. Besides, ANN uses L2-gradient to replace L1-gradient when doing back-propagation, which may lead to a wrong optimization direction. Using CNN rather than bigger ANN to distill can alleviate the problem.

*Q2: Compare with other self-distill methods.* A2: The experimental results using the proposed method and method of snapshot KD on CIFAR-10 is shown in Table 1 column 2 and 3, which shows the superiority of the proposed method.

Table 1: Compare with self-distill methods and dropout.

|  | PKKD | Snapshot KD | ANN + dropout |
|---|---|---|---|
| VGG-small | **95.03** | 93.95 | 93.88 |
| ResNet-20 | **92.96** | 92.33 | 92.20 |
| ResNet-32 | **93.62** | 93.17 | 93.09 |

Table 2: Operations in different networks.

| Model | Method | #Mul. | #Add. | XNOR |
|---|---|---|---|---|
|  | CNN | 1.8G | 1.8G | 0 |
| ResNet-18 | ANN | 0.1G | 3.5G | 0 |
|  | BNN | 0.1G | 1.8G | 1.7G |

*Q3: Moving experiment results to the main paper.* A3: The results will be placed in the main paper in final version.

**Response to R3:**

*Q1: Number of #Mul.* A1: Thank you for pointing out the problem. We follow the vanilla ANN setting [3] and '#Mul' was zero in that paper. The actual numbers are listed in Table 2. And we will fix it in the final version.

*Q2: Compare with other SOTA BNNs.* A2: The top-1 accuracies of PKKD-ANN and PCNN of ResNet-18 on ImageNet are **68.8** and 57.3. We will include these results in the final version.

*Q3: Extra training time.* A3: The training time of using traditional KD and PKKD for ResNet-18 on ImageNet is 40m 33s and 59m 45s per epoch.

Table 3: Compare with [R1], [25] and [26] on CIFAR-10.

|  | PKKD | [R1] | [25] | [26] |
|---|---|---|---|---|
| ResNet-20 | **92.96** | 92.38 | 92.22 | 92.27 |

Table 4: PKKD and KD in CNN distillation.

| ResNet-18 | 69.8/89.1 |
|---|---|
| PKKD | **73.1/91.3** |
| Traditional KD | 72.5/90.9 |

*Q4: Typos.* A4: We will fix all the typos in the final version.

**Response to R4:** All the following experiments and references will be included in the final version.

*Q1: Effectiveness of kernel method. Suggest comparing on ImageNet.* A1: To alleviate the influence of stochastic initialization, we have reported the mean accuracies of **5** different runs in Tab.1 in the main paper, which means the results are convincing. We will emphasize this in the final version. Results on ImageNet with ResNet-50 are shown in Table 5. We run 40 epochs due to the limited rebuttal period, but still shows the priority of PKKD.

Table 5: Ablation study on ImageNet with ResNet-50. 'K / NK' stands for using kernel or not. 'P / NP' stands for using progressive or fixed teacher.

| CNN | ANN | K + P (PKKD) | NK + P | K + NP | NK + NP |
|---|---|---|---|---|---|
| 73.2/91.6 | 70.4/89.9 | **73.4/91.7** | 72.5/90.9 | 72.0/90.6 | 71.3/90.4 |

*Q2: Compare with existing methods.* A2: We compare PKKD with [R1], [25] and [26] in Table 3 on CIFAR10. PKKD is better since it focus on alleviating the difference of distributions between CNN and ANN, which is the key problem.

*Q3: Reference FitNets. Difference from FitNets.* A3: We will reference FitNets in final version. The difference from FitNets is that FitNets use 1x1 Conv to make sure the size of teacher feature maps equal to student, so that the KD method can be applied correctly. In PKKD, the original size of feature maps from ANN and CNN are already the same. Thus, the purpose of using kernel (non-linear transformation, which is also different from FitNet) is to alleviate the difference of two distributions by mapping them to infinite dimensional space, and 1x1 Conv is further used to align the features in the new space derived from kernel. The difference has been briefly mentioned at the beginning of Section 3.1. More details will be included in the final version.

[R1] F. Tung G. Mori. Similarity-Preserving Knowledge Distillation. ICCV, 2019.

[Meta-Review · NeurIPS 2020]

I believe that by bridging the gap between Adder NN and CNNs this work provides a considerable contribution, allowing Adder NN to be considered among practical architecture and encouraging the community to research them further. In accordance with the reviewers, I think the proposed method is thoroughly investigated empirically. Please make sure to update the paper with all the results and answers that you have provided in your rebuttal.